# REM: From Structural Entropy To Community Structure Deception

**Yiwei Liu**[1,2]**, Jiamou Liu**[3]**, Zijian Zhang**[1,3]**, Liehuang Zhu**[1]*****, Angsheng Li**[4]

[1]School of Computer Science & Technology, Beijing Institute of Technology, Beijing 100081, China
[2]Institute of Cyberspace Research, Zhejiang University, Zhejiang 310027, China
[3]School of Computer Science, The University of Auckland, Auckland 1142, New Zealand
[4]School of Computer Science, Beihang University, Beijing 100083, China
{yiweiliu, zhangzijian, liehuangz}@bit.edu.cn, jiamou.liu@auckland.ac.nz, angsheng@buaa.edu.cn

## Abstract

This paper focuses on the privacy risks of disclosing the community structure in an online social network. By exploiting the community affiliations of user accounts, an attacker may infer sensitive user attributes. This raises the problem of community structure deception (CSD), which asks for ways to minimally modify the network so that a given community structure maximally hides itself from community detection algorithms. We investigate CSD through an information-theoretic lens. To this end, we propose a community-based structural entropy to express the amount of information revealed by a community structure. This notion allows us to devise residual entropy minimization (REM) as an efficient procedure to solve CSD. Experimental results over 9 real-world networks and 6 community detection algorithms show that REM is very effective in obfuscating the community structure as compared to other benchmark methods.

## 1 Introduction

Social networking sites facilitate effective communication through the means of Web feeds, discussion groups, timelines and more. Such a platform is characterized by a structure that consists of user accounts and their links. Discovering hidden patterns in this network structure is a compelling application of graph data mining algorithms. In particular, *community detection* stands out as one of the most important graph mining methods [11, 16, 23, 26]. Communities emerge as people naturally bond with those within the same working environment, family, or those who share similar tastes, interests and political viewpoints. By exploiting users' community affiliations, an attacker may infer certain personal – and sometimes sensitive – features of the users in a social network. For example, when the attacker has some background information asserting that several members of a community all work for the same organization. It is easy in this case to infer that other members of the same community also have ties with the organization. [29] showed that information about the community memberships of a user (i.e., the groups of a social network to which a user belongs) is sufficient to uniquely identify this person, or, at least, to significantly reduce the set of possible candidates. In [25], communities are used to re-identify multiple addresses belonging to a same user in Bitcoin trading networks. Therefore, there is a need to hide the community affiliations in order to preserve the privacy of online users.

This paper addresses the privacy risks due to community detection. Our goal is to minimally modify the network structure so that the community affiliations maximally hide themselves from community

detection algorithms. Despite growing interests on the privacy issues of social networks, very few works exist that target for community-level anonymization over a social network. Developing effective means for this problem faces numerous challenges: The first is the lack of a formal and universally-agreeable definition of communities. It is thus difficult to propose notions such as $k$-anonymity that are based on counting substructures and are independent from the community detection algorithms [32]. The second is the diversity of techniques used for community detection. An attacker may use many methods to identify communities in a network which makes it impossible to pinpoint a single objective metric that guides the deception of communities. The third is the desire to obfuscate not just a single community, but rather multiple communities or even all communities in the network. As opposed to existing work e.g. [28, 10] which focus on the deception of a single targeted community, we are interested in nullifying the community detection algorithms so that they are ineffective to identify any original communities in the obfuscated network.

This paper studies the *community structure deception problem (CSD)* that seeks a way to obfuscate a given community structure of a network through adding a fixed number of edges. (1) To solve this problem, we propose an information-theoretic perspective to this problem. This involves defining *community-based structural entropy* that captures the amount of information revealed by the community structure of a social network. (2) We propose a method to effectively nullify community detection algorithms based on the principle of *residual entropy minimization* (REM). REM clearly outperforms other schemes with the same goal which include a benchmark based on modularity minimization. (3) Our work derives new insights regarding structural entropy of a graph. These insights enable highly efficient implementation of our algorithm. (4) We experimentally validate the performance of our algorithm over 9 real-world networks and 6 community detection algorithms.

**Related work.** [2] showed that simply removing user identity is not sufficient to protect their privacy in an online social network. [31] systematically examined privacy threats in the online space. Efforts have been made to mitigate such risks on an *individual level*, i.e., identity leak [18, 13, 21, 33, 30, 7, 19], user attribute leak [5], social link disclosure [18, 13, 21, 33, 5], etc. Common structural obfuscation techniques include adding/removing edges, adding random noise, and contracting edges/nodes.

Community structure represents a way to partition vertices of a complex network into dense subgraphs that are sparsely connected among each other [11]. Many community detection algorithms exist, e.g., Louvain method that utilizes modularity is commonly used [3]. Recent years, several works that address the problem of hiding a given community in a network have emerged. E.g. [20] aimed to hide a community by adding edges. They only considered a modularity-based community detection algorithm. [28, 10] studied this problem by rewiring edges.

Quantifying structural information is an important challenge in information theory. [24] proposed the first entropy measure for graphs. This is followed by several other notions such as parametric graph entropy [8], Gibbs entropy [1], Shannon entropy and Von Neumann entropy [4]. All of these measures are simply the Shannon entropy applied to different types of distributions. Based on the idea of random walks, the entropy defined in [26] determines the average number of bits per step by using the ergodic node visit frequencies on a network. After that, [17] defined the structure entropy of a graph as the minimum numbers of bits to encode the vertex that is accessible from a step of random walk. In this paper, we follow these ideas and utilize similar notions for community structure deception.

## 2 Problem Formulation

We model a social network as an undirected connected graph $G = (V, E)$, where $V$ is a set of vertices which represent user accounts and $E$ is a set of edges of the form $\{u, v\} \subseteq V$ ($u \neq v$) which represent social ties. The *volume* of any $U \subseteq V$ is the sum of the *degrees* $d_v$ of all $v \in U$. The *community structure* of $G$ refers to a partition $\mathcal{P}$ of $V$. More formally, $\mathcal{P}$ is an equivalence relation over the set of vertices $V$ whose equivalence classes are called *communities*. If $i$ and $j$ in the same community, we write $(i, j) \in \mathcal{P}$. We assume that the input to our problem consists of a social network $G$ and a *community structure* $\mathcal{P}$ to be obfuscated. This community structure $\mathcal{P}$ is characterized by high internal density and low external density. For convenience, we sometimes abuse the notation representing $\mathcal{P}$ also as the collection of its equivalence classes $\{X_1, X_2, \ldots, X_L\}$ where $L \in \mathbb{N}$ and each $X_i$ is a community; $\nu_i$ denotes the volume of $X_i$ and $g_i$ denotes the number of edges with

exactly one end point in $X_i$. The following hypothesis lays down the fundamental assumption of the community deception problem:

**Hypothesis 1** (Community deception hypothesis). *The disclosure of the community structure $\mathcal{P}$ of a graph $G = (V, E)$ leads to privacy leak and should be avoided.*

Given $G = (V, E)$, a *community detector* $\mathscr{F}$ is a procedure that reveals an equivalence relation $\mathscr{F}(G)$ over $V$ to resemble the ground truth community structure $\mathcal{P}$. Hypothesis 1 asserts the necessity of distorting the network data $G$, so that no community detector $\mathscr{F}$ will truthfully report the original community structure $\mathcal{P}$. In this paper, we focus on network distortions as a result of adding a number of "dummy edges" between unconnected vertices in the network.

**Definition 1.** *For $G = (V, E)$ and a set $E'$, an* edge expansion *is a graph $G \oplus E' := (V, E \cup E')$.*[2]

Given $G = (V, E)$ and a community structure $\mathcal{P}$ on $G$, a *community structure deceptor* produces an edge expansion $G'$ of $G$ so that any community detection algorithm $\mathscr{F}$ is nullified on $G'$. The precise definition relies on what it means for the algorithm $\mathscr{F}$ to be "nullified on $G'$". Several narratives exist for this phrase. Suppose $\mathcal{P}'$ is the community structure $\mathscr{F}(G')$ output by $\mathscr{F}$. The first narrative asserts that $\mathcal{P}'$ is dissimilar with $\mathcal{P}$. The second asserts that very little information is revealed about $\mathcal{P}$ from $\mathcal{P}'$. The third states that $\mathscr{F}$ is ineffective in answering same-community queries.

**Narrative 1: Partition similarity.** One may apply a standard set-based metric, *Jaccard index*, to compare $\mathcal{P}$ and $\mathcal{P}'$: Set $J(\mathcal{P}, \mathcal{P}') := |\mathcal{P} \cap \mathcal{P}'|/|\mathcal{P} \cup \mathcal{P}'|$ (treating $\mathcal{P}$ and $\mathcal{P}'$ as relations); We adopt $J(\mathcal{P}, \mathcal{P}')$ for its simplicity and correlation with other measures, e.g., *transfer distance* of $\mathcal{P}$ & $\mathcal{P}'$ [9]. A good community structure deceptor should return $\mathcal{P}'$ with small $J(\mathcal{P}, \mathcal{P}')$.

**Narrative 2: Mutual information.** *Normalized mutual information* (NMI) measures the amount of common information between two random variables. Take community structures $\mathcal{P} = \{X_1, \ldots, X_p\}$ and $\mathcal{P}' = \{X'_1, \ldots, X'_q\}$. Define

$$H(\mathcal{P}) = -\sum_{i=1}^{p} \frac{|X_i|}{|V|} \log \frac{|X_i|}{|V|}, \text{ and } H(\mathcal{P}|\mathcal{P}') = -\sum_{i=1}^{p}\sum_{j=1}^{q} \frac{|X_i \cap X'_j|}{|V|} \log \frac{|X_i \cap X'_j|/|V|}{|X'_j|/|V|}.$$

Mutual information is then defined as $I(\mathcal{P}, \mathcal{P}') := H(\mathcal{P}) - H(\mathcal{P}|\mathcal{P}')$. NMI is thus $D(\mathcal{P}, \mathcal{P}') = \frac{I(\mathcal{P}, \mathcal{P}')}{\max\{H(\mathcal{P}), H(\mathcal{P}')\}}$ [14]. $D$ satisfies both the normalization and the metric properties, and utilizes the range $[0, 1]$ well [27]. A community structure deceptor should return $\mathcal{P}'$ with small $D(\mathcal{P}, \mathcal{P}')$.

**Narrative 3: Query accuracy.** One may imagine that the community detection algorithm $\mathscr{F}$ facilitates an adversary who aims to perform *same-community queries* about user accounts. This query returns true for any distinct $i, j \in V$ if $(i, j) \in \mathcal{P}'$ and false otherwise. The *recall* of this query is

$$R(\mathcal{P}, \mathcal{P}') = \frac{|\{(i, j) \in \mathcal{P} \mid i \neq j, (i, j) \in \mathcal{P}'\}|}{|\{(i, j) \in \mathcal{P} \mid i \neq j\}|}.$$

A procedure that returns true for any pair of vertices $(i, j)$ with probability $1/2$ has an expected recall of $50\%$. Hence $\mathscr{F}$ can be considered nullified when $R(\mathcal{P}, \mathcal{P}') \leq 50\%$.

The *community structure deception* (CSD) problem is defined as demanding a community structure deceptor for a given network $G$ with its community structure $\mathcal{P}$. Furthermore, the deceptor should add a bounded budget $k \in \mathbb{N}$ of edges to $G$ in the hope to get the best deception effect. One initial idea to solve CSD is to fix a community detector $\mathscr{F}$ and $S \in \{J, D, R\}$, and to solve the problem

$$\text{minimize } S(\mathcal{P}, \mathscr{F}(G \oplus E')) \quad \text{subject to } |E'| \leq k.$$

There are several reasons why this would not be a good approach: (1) The functions $J$, $D$ and $R$ all depend on the output of the algorithm $\mathscr{F}$; however the CSD problem demands the obfuscation of the community structure $\mathcal{P}$ regardless of how communities are detected. (2) Choosing any one of $J$, $D$ and $R$ leads only to optimizing a single criterion. (3) Solving the optimization problem may require examining all $k$-tuples of potential edges which leads to prohibitive time cost.

A more reasonable approach is to identify a uniform criterion which is independent of how communities are detected. One natural candidate for such a metric is *modularity*. *Modularity* of $\mathcal{P}$ measures

the difference between the density of its communities and the expected density of a null model [22]:

$$M_{\mathcal{P}}(G) = \sum_{i=1}^{L} \left[ \frac{\nu_i - g_i}{2|E|} - \left( \frac{\nu_i}{2|E|} \right)^2 \right]. \tag{1}$$

*Modularity maximization* has been a widely-used principle for community detection. In general, a large $\max M_{\mathcal{P}}(G)$ implies the existence of a prominent community structure in $G$. To obfuscate the community structure, it thus makes sense – at least in principle – to minimize the modularity $M_{\mathcal{P}}(G)$.

**Definition 2.** *A* modularity minimizing (MOM) deceptor *is an algorithm that outputs an edge $e$ such that the modularity $M_{\mathcal{P}}(G \oplus e)$ is minimized.*

In actual fact, however, MOM deceptor is not a good choice for CSD : Firstly, it is not hard to prove that, the MOM deceptor will always try to create edges between two communities $X_i$, $X_j$ in $\mathcal{P}$ with the largest combined volume $\nu_i + \nu_j$. Therefore $k$ edges created by iterations of MOM will most likely affect only two communities, and the obfuscated network will not hide $\mathcal{P}$ effectively. Secondly, modularity's significance primarily lies in identifying the most prominent community structure, i.e., the one that maximizes modularity. The MOM deceptor, on the other hand, concerns with modularity of a given partition $\mathcal{P}$ which may not be modularity maximizing. Thirdly, modularity sometimes fails for its purpose since a random graph – a structure that does not exhibit a clear community structure – may also have partitions with large modularity [12]. These limitations calls for a new method for CSD.

## 3  REM**: Residual Entropy-based CSD**

To derive a solution for CSD that is independent of the community detector, it makes sense to inquire the information content of a community structure $\mathcal{P}$ in $G$. Imagine $G$ as a network where vertices are able to pass messages through edges. The delivery of a message from a sender $u$ to a *receiver $v$*, where $\{u, v\} \in E$, is named a *call*. Intuitively, a call is one directed flow of message. Therefore an undirected edge $\{u, v\}$ allows messages to be passed in both directions. Now imagine, to explore $G$, an exogenous process continuously collects such *calls* uniformly at random. This differs from the random walk in [26] where the receiver of a call is the sender of the next call. Hence, at any moment, the probability that $v$ is a call's receiver is $d_v/(2|E|)$. We are interested in an encoding of vertices of the network based on this probability distribution [17].

**Definition 3.** Structural entropy $\mathcal{H}(G)$ *captures the average number of bits needed to encode the receivers of the calls in a lossless way: $\mathcal{H}(G)$ equals Shannon's entropy of the distribution $(d_i/2|E|)_{i \in V}$, i.e.,*

$$\mathcal{H}(G) := -\sum_{i=1}^{|V|} \frac{d_i}{2|E|} \log_2 \frac{d_i}{2|E|}. \tag{2}$$

$\mathcal{H}(G)$ merely expresses the average information of a call in $G$ *without* assuming any community structure. Now assume the presence of $\mathcal{P} = \{X_1, X_2, \ldots, X_L\}$. The structural information of a community $X_j$ consists of two levels: (a) looking from a vertex level, the information of any vertex $i \in X_j$ as a receiver of messages, and (b) looking from a community level, the information of the entire community $X_j$ as a receiver of messages. These two levels of meanings can be reflected in (2) through the equation $-\log_2 \frac{d_i}{2|E|} = -\log_2 \frac{d_i}{\nu_j} - \log_2 \frac{\nu_j}{2|E|}$. The first term above is the average numbers of bits necessary to describe $i$ in $X_j$ and the second term is the average numbers of bits to describe the community $X_j$. We once again assume an exogenous process that continuously collects calls between vertices in $G$ u.a.r., but with the following difference: Since we are given the community structure $\mathcal{P} = \{X_1, X_2, \ldots, X_L\}$, we can omit the community level codeword if the sender $u$ and receiver $v$ belong to the same community. Thus, when encoding a vertex $i \in X_j$, just like above, we encode $i$ at the vertex level as $-\log_2 \frac{d_i}{\nu_j}$ and then encode $X_j$ at the community level as $-\log_2 \frac{\nu_j}{2|E|}$. For (a), the information for all vertices in $X_j$ as receivers is $\mathcal{H}(G \restriction_{X_j})$ with the probability $\frac{\nu_j}{2|E|}$, where $\mathcal{H}(G \restriction_{X_j}) := -\sum_{i \in X_j} \frac{d_i}{\nu_j} \log_2 \frac{d_i}{\nu_j}$. For (b), the information for $X_j$ as a receiver is $-\log_2 \frac{\nu_i}{|2E|}$ with the probability $\frac{g_j}{2|E|}$ since in this case we only consider calls whose senders are not in $X_j$. The expected information gives us the following structural entropy measure [17]:

**Definition 4.** *The* structural entropy of $G$ relative to $\mathcal{P}$ is

$$\mathcal{H}_{\mathcal{P}}(G) := \sum_{j=1}^{L} \left[ \frac{\nu_j}{2|E|} \mathcal{H}\left(G \upharpoonright_{X_j}\right) - \frac{g_j}{2|E|} \log_2 \frac{\nu_j}{2|E|} \right] \tag{3}$$

Note that $\mathcal{H}(G) = \mathcal{H}_{\mathcal{P}}(G)$ when $\mathcal{P}$ is either the trivial partition that puts all vertices in the same community, or the partition where each community is a singleton. We thus view both $\mathcal{H}(G)$ and $\mathcal{H}_{\mathcal{P}}(G)$ as expressing states of the community structure. $\mathcal{H}(G)$ expresses the entropy of $G$ in a basic "reference partition", and $\mathcal{H}_{\mathcal{P}}$ reflects the effect of enforcing partition $\mathcal{P}$ on $G$. Their difference thus measures the gained amount of certainty as the communities in $\mathcal{P}$ take shape.

**Definition 5.** *The* normalized residual entropy of $\mathcal{P}$ is

$$\rho_{\mathcal{P}}(G) := (\mathcal{H}(G) - \mathcal{H}_{\mathcal{P}}(G))/\mathcal{H}(G). \tag{4}$$

In principle, a smaller $\rho_{\mathcal{P}}(G)$ means that $\mathcal{P}$ contains less information about $G$ and thus is harder to detect. To hide the communities in $\mathcal{P}$, it thus makes sense to reduce the residual entropy through modifying the network structure.

**Definition 6.** *A* residual entropy minimizing (REM) deceptor *is an algorithm that outputs an edge $e$ such that the normalized residual entropy $\rho_{\mathcal{P}}(G \oplus e)$ is minimized.*

A crude implementation of an REM deceptor examines each potential edge $e$ that is missing from the current graph and compares $\rho_{\mathcal{P}}(G \oplus e)$. This implementation runs in $\Omega(|V|^2)$ time, rendering itself inapplicable for large graphs. We instead present an $O(L|V|)$-implementation where $L$ is the number of communities $X_1, \ldots, X_L$ in $\mathcal{P}$. This is a much more efficient implementation assuming $L \ll |V|$.

Take $s, t \in \{1, \ldots, L\}$. A *non-edge* is a pair $\{u, v\} \notin E$ with $(u, v) \in X_s \times X_t$; the *volume* of this non-edge is $d_u + d_v$. Assume $X_s \times X_t$ contains a non-edge. Let $\delta_{s,t}$ be the smallest degree of any vertex $v \in X_s \cup X_t$ that is in a non-edge. Let $\beta_{s,t}$ be the smallest volume of any non-edge $\{u, v\}$ with $\min\{d_u, d_v\} = \delta_{s,t}$. A non-edge $\{u, v\}$ is called *critical* if its volume $d \leq \beta_{s,t}$ and $\min\{d_u, d_v\}$ is the smallest among all non-edges with volume $d$. Algorithm 1 presents our REM deceptor from the following lemmas.

---

**Algorithm 1:** An efficient REM deceptor

---

**Input:** Graph $G = (V, E)$, $\mathcal{P} = \{X_1, X_2, \ldots, X_L\}$
**Output:** A non-edge $\{u^*, v^*\}$
1 Initialize $\rho^* \leftarrow 1$;
2 **for** $s \leftarrow 1$ *to* $L$ *and* $t \leftarrow s$ *to* $L$ **do**
3      **if** $X_s \times X_t$ *contains no non-edge* **then**
4          continue;
5      **for** *all critical non-edge* $\{u, v\}$ *in* $X_s \times X_t$ **do**
6          Set $\rho_{u,v} \leftarrow (\mathcal{H}(G \oplus \{u, v\}) - \mathcal{H}_{\mathcal{P}}(G \oplus \{u, v\}))/\mathcal{H}(G \oplus \{u, v\})$;
7          **if** $\rho_{u,v} < \rho^*$ **then**
8              Set $\rho^* \leftarrow \rho_{u,v}$, $u^* \leftarrow u$, and $v^* \leftarrow v$;

9 return $\{u^*, v^*\}$;

---

**Lemma 1.** *For non-edges $\{u, v\}, \{x, y\}$, $\min\{d_u, d_v\} \leq \min\{d_x, d_y\}$ and $d_u + d_v \leq d_x + d_y$ implies that $\mathcal{H}(G \oplus \{u, v\}) \geq \mathcal{H}(G \oplus \{x, y\})$.*

*Proof.* Define the function $F \colon \mathbb{R} \to \mathbb{R}$ by $F(x) = \frac{(x+1)\log_2(x+1) - x\log_2(x)}{2(|E|+1)}$. We remark that the function $F$ is *monotonic*, as $F'(x) = \frac{\log_2(x+1) - \log_2(x)}{2(|E|+1)} > 0$, and is *convex* for $x > 0$, as $F''(x) = -\frac{1}{2\ln 2 \cdot (|E|+1)(x+1)x} < 0$. Moreover, the following hold:

$$\forall u, v, w \in V: \mathcal{H}(G \oplus \{u, v\}) - \mathcal{H}(G \oplus \{u, w\}) = F(d_w) - F(d_v), \quad \text{and thus}$$

$$\forall u, v, x, y \in V: \mathcal{H}(G \oplus \{u, v\}) - \mathcal{H}(G \oplus \{x, y\})$$

$$= (\mathcal{H}(G \oplus \{u, v\}) - \mathcal{H}(G \oplus \{v, y\})) + (\mathcal{H}(G \oplus \{v, y\}) - \mathcal{H}(G \oplus \{x, y\}))$$

$$= (F(d_y) - F(d_v)) + (F(d_x) - F(d_u)) \tag{5}$$

Assume w.l.o.g. $d_v \leq d_u$, $d_y \leq d_x$ and $d_v \leq d_y$. If $d_u \leq d_x$, then by (5) and monotonicity of $F$, we have $\mathcal{H}(G \oplus \{u,v\}) \geq \mathcal{H}(G \oplus \{x,y\})$. Now suppose $d_x < d_u$. Then $d_v < d_y \leq d_x < d_u$. By Lagrange's mean value theorem, there exist $d_x < \xi < d_u$ and $d_v < \zeta < d_y$ such that

$$\frac{F(d_u) - F(d_x)}{d_u - d_x} = F'(\xi) < F'(\zeta) = \frac{F(d_y) - F(d_v)}{d_y - d_v}, \tag{6}$$

where the inequality is due to convexity of $F$. Since $d_u - d_x \leq d_y - d_v$, we have $F(d_u) - F(d_x) < F(d_y) - F(d_v)$. The lemma then follows from (5). $\square$

**Lemma 2.** *For any two communities $X_i$, $X_j$ in $\mathcal{P}$, any non-edges $e_1, e_2$ whose endpoints link $X_i$ and $X_j$, we have $\mathcal{H}(G \oplus e_1) - \mathcal{H}_\mathcal{P}(G \oplus e_1) = \mathcal{H}(G \oplus e_2) - \mathcal{H}_\mathcal{P}(G \oplus e_2)$.*

*Proof.* Define distributions $Y \sim (\frac{\nu_1}{2|E|}, \ldots, \frac{\nu_L}{2|E|})$ and $Z \sim (\frac{c_{1,1}}{\nu_1}, \ldots, \frac{c_{n,1}}{\nu_1}, \ldots, \frac{c_{1,L}}{\nu_L}, \ldots, \frac{c_{n,L}}{\nu_L})$ where $c_{i,j} = d_i$ if $i \in X_j$, and $c_{i,j} = 0$ otherwise. The entropy of the joint probability is

$$H(Y, Z) = -\sum_{i=1}^{n} \frac{d_i}{2|E|} \log_2 \frac{d_i}{2|E|} = \mathcal{H}(G) \tag{7}$$

By (7) and the chain rule (see e.g.[6]),

$$\begin{aligned}
H(Y, Z) &= H(Z|Y) + H(Y) = \sum_{j=1}^{L} \frac{\nu_j}{2|E|} H(Z|Y = j) + H(Y) \\
&= \sum_{j=1}^{L} \left[ \frac{\nu_j}{2|E|} \mathcal{H}\left(G \upharpoonright_{X_j}\right) - \frac{\nu_j}{2|E|} \log_2 \frac{\nu_j}{2|E|} \right]
\end{aligned}$$

The following can then be obtained from (3):

$$\mathcal{H}(G) - \mathcal{H}_\mathcal{P}(G) = -\sum_{j=1}^{L} \frac{\nu_j - g_j}{2|E|} \log_2 \frac{\nu_j}{2|E|} \tag{8}$$

The lemma follows from (8) as $G \oplus e_1$ and $G \oplus e_2$ have the same values of $\nu_j$ and $g_j$. $\square$

A non-edge $e$ is *RE-minimizing* if $\rho_\mathcal{P}(G \oplus e)$ is the smallest among all non-edges. The next lemma states that, to find an RE-minimizing non-edge, an REM deceptor only needs to consider critical non-edges.

**Lemma 3.** *There exists a critical non-edge $\{u, v\} \in X_i \times X_j$ for some $i, j$ that is RE-minimizing.*

*Proof.* Take an RE-minimizing non-edge $e_1 = \{x, y\}$ and say $x \in X_s, y \in X_t$. Suppose $e_1$ is not critical. There are two cases: Firstly, if $d_x + d_y > \beta_{s,t}$, let $e_2 = \{u, v\}$ be the critical non-edge with $(u, v) \in X_s \times X_t$ and $\min\{d_u, d_v\} = \delta_{s,t} \leq \min\{d_x, d_y\}$. By Lem. 1, $\mathcal{H}(G \oplus e_2) \geq \mathcal{H}(G \oplus e_1)$. Secondly, if $d_x + d_y \leq \beta_{s,t}$, then $\min\{d_u, d_v\} < \min\{d_x, d_y\}$ for some critical non-edge $e_2 = \{u, v\}$ between $X_s$ and $X_t$ with the same volume $d_x + d_y$. In this case, we still have $\mathcal{H}(G \oplus e_2) \geq \mathcal{H}(G \oplus e_1)$. In either case, Lem. 2 asserts that $\mathcal{H}(G \oplus e_1) - \mathcal{H}_\mathcal{P}(G \oplus e_1) = \mathcal{H}(G \oplus e_2) - \mathcal{H}_\mathcal{P}(G \oplus e_2)$. Thus by (4), $\rho_\mathcal{P}(G \oplus e_2) \leq \rho_\mathcal{P}(G \oplus e_1)$ and $e_2$ is critical. $\square$

**Theorem 2.** *Alg. 1 implements* REM *deceptor in $O(L|V|)$.*

*Proof.* The Alg. 1 goes over all $s, t \in \{1, \ldots, L\}$ and critical non-edges $e = \{u, v\}$ in $X_s \times X_t$ to find a critical non-edge $\{u, v\}$ that minimizes $\rho_\mathcal{P}(G \oplus e)$. By Lemma 3, $\{u, v\}$ is RE-minimizing.

For communities $X_s$ and $X_t$, suppose a data structure is used that assigns to each node $x \in X_s$ and the node $x' \in X_t$ where no edge exists between $x$ and $x'$, and $x'$ is such a node with minimum degree. To find the desired critical edge, the algorithm may scan over all such pairs $(x, x')$, where $x \in X_s$. This takes $O(|X_s|)$. Similarly, the algorithm examines over all pairs $(y, y')$ where $y \in X_t$ and $y' \in X_s$ is defined analogously as $x'$. This takes $O(|X_t|)$. Hence, for $X_s$ and $X_t$ the algorithm takes $O(|X_s + X_t|)$. Thus, for any $X_s$, the algorithm will take $O(L|X_s| + |X_1| + \cdots + |X_L|) = O(L|X_s| + |V|)$. The overall time takes $O(L|X_1| + \cdots + L|X_L| + L|V|) = O(L|V|)$. The

implementation of the required data structure would store for each node $x \in X_s$, collections of nodes $Y_d, Y_{d+1}, \ldots, Y_g \subset X_j$ where $Y_d$ contains all nodes $y \in X_t$ such that $\{x, y\}$ is a non-edge and $y$ has degree $d$, where $d, g$ are least and greatest integers where $Y_d, Y_g$ are non-empty. This makes sure that the data structure can be built and updated in the required time complexity.

$\square$

In real-world networks, the links between two communities are sparse and usually vertices with the smallest degree in each community are not linked. In this case, any critical non-edge $\{u, v\}$ in $X_s \times X_t$ satisfies $d_u = \min_{x \in X_s} d_x$ and $d_v = \min_{y \in X_t} d_y$. The algorithm takes only $O(L^2)$ when the vertices with the smallest degree in each community are given.

## 4 Experiments

**Dataset.** We evaluate the performance of our algorithm over 9 real-world networks from [`http://konect.uni-koblenz.de/`]. The networks are chosen from a range of domains, including human contacts: jazz (Jaz); animal network: dolphin (Dol); communication network: email (Eml), pretty good privacy (PGP); infrastructure network: powergrid (Pow); computer network: CAIDA (CAI); and online networks: Facebook (Fbk), Brightkite (Bri), Livemocha (Liv). Due to the limitations of the efficiency of some community detectors $\mathscr{F}$, we do not select a particularly large dataset. In fact, our REM deceptor can handle large datasets according to the complexity analysis in Theorem 2. To validate the efficiency of REM, we list the running time of applying Alg. 1 for one edge. This is compared with a 'crude' implementation of REM deceptor which resembles Alg. 1, instead of examining only the critical non-edges, goes over *all* non-edges in $G$ to look for the RE-minimizing one[3]. See details in Table 1.

Table 1: Specifics of the datasets , the number of communities and running time.

| Dataset | $|V|$ | $|E|$ | Number of communities | | | | | | 'crude' (ms) | REM (ms) |
| --- | --- | --- | --- | --- | --- | --- | --- | --- | --- | --- |
| | | | btw | gre | inf | lou | spi | wal | | |
| Dol | 62 | 159 | 4 | 4 | 5 | 5 | 5 | 4 | 13.7 | 0.305 |
| Jaz | 198 | 2,741 | 12 | 4 | 7 | 4 | 4 | 11 | 74.2 | 0.296 |
| Eml | 1,133 | 5,451 | 11 | 16 | 70 | 10 | 13 | 49 | 3960 | 2.40 |
| Fbk | 2,888 | 2,981 | 8 | 8 | 11 | 8 | 11 | 6 | 26,300 | 5.67 |
| Pow | 4,941 | 6,594 | 45 | 43 | 486 | 43 | 25 | 364 | 66,300 | 8.67 |
| PGP | 10,680 | 24,316 | - | 189 | 1066 | 96 | 25 | 1574 | 5mins | 29.8 |
| CAI | 26,475 | 53,381 | - | 44 | 1382 | 38 | - | 667 | 4hrs | 28.6 |
| Bri | 58,228 | 214,078 | - | 1682 | 4813 | 687 | - | 6892 | 5.5hrs | 1010 |
| Liv | 104,103 | 2,193,083 | - | 189 | - | 14 | - | - | > 1day | 179 |

**Community detectors.** An adversary attacks by applying a community detector $\mathscr{F}$. For this we use six well-known algorithms [11]: ▷ Edge-Betweeness(btw) is a hierarchical decomposition process where edges are removed in decreasing order of their edge betweenness scores and runs in $O(|E|^2|V|)$. ▷ Greedy(gre) is a greedy modularity maximization strategy and runs in $O(|V| \log^2 |V|)$. ▷ InfoMap(inf) detects communities that have the shortest description length for a random walk in $O(|E|)$. ▷ Louvain(lou) is a multi-level modularity optimization algorithm which runs in $O(|V| \log |V|)$. ▷ SpinGlass(spi) finds communities by searching for the ground state of an infinite spin glass and runs in $O(|V|^3)$. ▷ WalkTrap(wal) detects communities using random walks and runs in $O(|V|^2 \log |V|)$. Table 1 shows the number of communities found by each algorithm. If the algorithm does not terminate in 2 hours on a dataset, a '-' is written in the table.

**Community structure deceptors.** We compare REM with two other CS deceptors, including MOM and a benchmark RAN that adds randomly chosen non-edges. The experiments aim to: (1) check if the normalized residual entropy correlates with the indices $J, D, R$ (in Sec. 2); (2) compare the effectiveness of the deceptors in hiding community structures; (3) discuss the preservation of data's key indicators after applying REM. To this end, we ran the experiments in the worst-case scenario that the initial community structure $\mathcal{P}$ is fully detected by community detector $\mathscr{F}$. We then apply the deceptors MOM, REM and RAN to obfuscate the network $G$ and apply $\mathscr{F}$ on the obfuscated

network $G \oplus E'$. The indices $\{J, D, R\}$ are calculated for $\mathcal{P}$ and the new structure $\mathcal{P}'$, where $\mathcal{P}' = \mathscr{F}(G \oplus E')$. Each value in the figures and tables is the average of 30 runs.

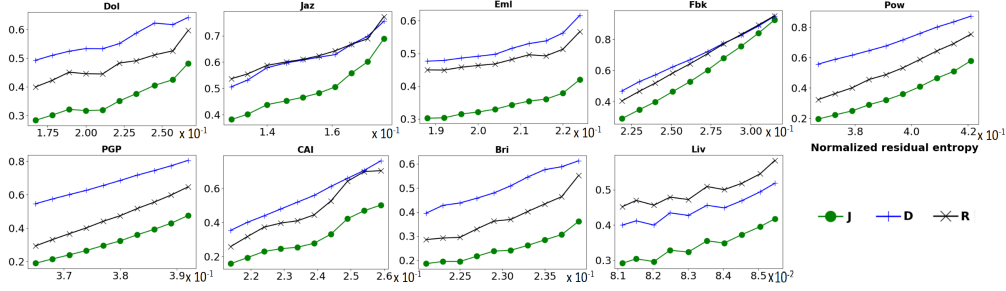

Figure 1: The trend of $J$, $D$ and $R$ with the community normalized residual entropy in our datasets.

Table 2: The $J$, $D$ and $R$ indices based on different deceptors for 6 kinds of community detectors.

**Edge-Betweeness**

| Data | Jaccard(J) | | | NMI(D) | | | Recall(R) | | |
|---|---|---|---|---|---|---|---|---|---|
| | RAN | MOM | REM | RAN | MOM | REM | RAN | MOM | REM |
| Dol | 0.63 | 0.55 | **0.47** | 0.63 | 0.55 | **0.43** | 0.86 | 0.76 | **0.66** |
| Jaz | 0.56 | 0.60 | **0.32** | **0.37** | 0.57 | 0.39 | 0.98 | 1.00 | **0.41** |
| Eml | 0.90 | 0.94 | **0.46** | 0.24 | 0.69 | **0.09** | 0.97 | 0.99 | **0.47** |
| Fbk | 0.43 | 0.58 | **0.31** | 0.51 | 0.89 | **0.41** | **0.58** | 0.99 | 0.60 |
| Pow | 0.23 | 0.69 | **0.13** | | | **0.49** | 0.39 | 0.87 | **0.28** |

**Greedy**

| Data | Jaccard(J) | | | NMI(D) | | | Recall(R) | | |
|---|---|---|---|---|---|---|---|---|---|
| | RAN | MOM | REM | RAN | MOM | REM | RAN | MOM | REM |
| Dol | 0.58 | 0.50 | **0.44** | 0.68 | 0.65 | **0.55** | 0.65 | 0.58 | **0.51** |
| Jaz | 0.69 | 0.51 | **0.38** | 0.68 | 0.60 | **0.35** | 0.83 | 0.72 | **0.62** |
| Eml | 0.32 | 0.30 | **0.26** | 0.38 | 0.43 | **0.33** | 0.44 | 0.42 | **0.38** |
| Fbk | 0.45 | 0.46 | **0.29** | 0.53 | 0.72 | **0.45** | 0.55 | 0.53 | **0.41** |
| Pow | 0.25 | 0.62 | **0.19** | 0.60 | 0.92 | **0.54** | 0.39 | 0.88 | **0.32** |
| PGP | 0.38 | 0.52 | **0.25** | 0.64 | 0.89 | **0.59** | 0.51 | 0.70 | **0.36** |
| CAI | 0.23 | 0.39 | **0.17** | 0.38 | 0.72 | **0.32** | 0.35 | 0.58 | **0.29** |
| Bri | 0.44 | **0.20** | 0.28 | 0.56 | **0.52** | 0.54 | 0.58 | **0.32** | 0.37 |

**InfoMap**

| Data | Jaccard(J) | | | NMI(D) | | | Recall(R) | | |
|---|---|---|---|---|---|---|---|---|---|
| | RAN | MOM | REM | RAN | MOM | REM | RAN | MOM | REM |
| Dol | 0.65 | 0.51 | **0.47** | 0.78 | 0.72 | **0.64** | 0.74 | 0.60 | **0.54** |
| Jaz | 0.52 | 0.60 | **0.48** | **0.05** | 0.52 | 0.06 | 1.00 | 1.00 | **0.88** |
| Eml | 0.47 | 0.63 | **0.34** | 0.77 | 0.92 | **0.71** | 0.61 | 0.94 | **0.44** |
| Fbk | 0.27 | 0.32 | **0.09** | 0.46 | 0.57 | **0.37** | 0.27 | 0.32 | **0.10** |
| Pow | 0.47 | 0.75 | **0.33** | 0.91 | 0.97 | **0.87** | 0.61 | 0.87 | **0.48** |
| PGP | 0.61 | 0.81 | **0.39** | 0.93 | 0.99 | **0.88** | 0.72 | 0.94 | **0.49** |
| CAI | 0.24 | 0.83 | **0.18** | 0.77 | 0.98 | **0.74** | 0.27 | 0.98 | **0.22** |

**Louvain**

| Data | Jaccard(J) | | | NMI(D) | | | Recall(R) | | |
|---|---|---|---|---|---|---|---|---|---|
| | RAN | MOM | REM | RAN | MOM | REM | RAN | MOM | REM |
| Dol | 0.62 | 0.55 | **0.41** | 0.74 | 0.73 | **0.62** | 0.75 | 0.65 | **0.52** |
| Jaz | 0.72 | 0.52 | **0.38** | 0.74 | 0.62 | **0.51** | 0.84 | 0.67 | **0.54** |
| Eml | 0.37 | 0.42 | **0.30** | 0.54 | 0.67 | **0.48** | 0.52 | 0.72 | **0.45** |
| Fbk | 0.43 | 0.43 | **0.29** | 0.50 | 0.70 | **0.47** | 0.52 | 0.47 | **0.40** |
| Pow | 0.24 | 0.60 | **0.19** | 0.62 | 0.90 | **0.56** | 0.38 | 0.86 | **0.32** |
| PGP | 0.28 | 0.53 | **0.19** | 0.62 | 0.90 | **0.55** | 0.42 | 0.77 | **0.29** |
| CAI | 0.21 | 0.30 | **0.16** | 0.40 | 0.71 | **0.35** | 0.32 | 0.43 | **0.26** |
| Bri | 0.35 | 0.32 | **0.26** | 0.58 | 0.70 | **0.54** | 0.52 | 0.48 | **0.40** |
| Liv | 0.56 | 0.52 | **0.45** | 0.61 | 0.57 | **0.54** | 0.72 | 0.69 | **0.59** |

**SpinGlass**

| Data | Jaccard(J) | | | NMI(D) | | | Recall(R) | | |
|---|---|---|---|---|---|---|---|---|---|
| | RAN | MOM | REM | RAN | MOM | REM | RAN | MOM | REM |
| Dol | 0.73 | 0.55 | **0.47** | 0.82 | 0.75 | **0.65** | 0.82 | 0.64 | **0.56** |
| Jaz | 0.69 | 0.49 | **0.39** | 0.70 | 0.57 | **0.52** | 0.86 | 0.61 | **0.54** |
| Eml | 0.44 | 0.56 | **0.26** | 0.63 | 0.81 | **0.55** | 0.61 | 0.81 | **0.49** |
| Fbk | 0.44 | 0.42 | **0.30** | 0.53 | 0.70 | **0.49** | 0.52 | 0.48 | **0.40** |
| Pow | 0.20 | 0.38 | **0.19** | 0.53 | 0.74 | **0.49** | 0.32 | 0.59 | **0.31** |
| PGP | 0.21 | 0.28 | **0.13** | 0.44 | 0.53 | **0.32** | 0.33 | 0.44 | **0.22** |

**WalkTrap**

| Data | Jaccard(J) | | | NMI(D) | | | Recall(R) | | |
|---|---|---|---|---|---|---|---|---|---|
| | RAN | MOM | REM | RAN | MOM | REM | RAN | MOM | REM |
| Dol | 0.45 | 0.39 | **0.35** | 0.63 | 0.58 | **0.57** | 0.51 | 0.47 | **0.37** |
| Jaz | 0.67 | 0.52 | **0.43** | 0.70 | 0.65 | **0.62** | 0.82 | 0.99 | **0.54** |
| Eml | 0.37 | 0.59 | **0.30** | **0.61** | 0.89 | 0.64 | 0.53 | 0.96 | **0.45** |
| Fbk | 0.28 | 0.24 a | **0.23** | 0.24 | 0.30 | **0.22** | 0.41 | **0.30** | 0.20 |
| Pow | 0.11 | 0.45 | **0.08** | 0.77 | 0.94 | **0.73** | 0.26 | 0.95 | **0.20** |
| PGP | 0.40 | 0.66 | **0.18** | 0.91 | 0.99 | **0.86** | 0.52 | 0.98 | **0.26** |

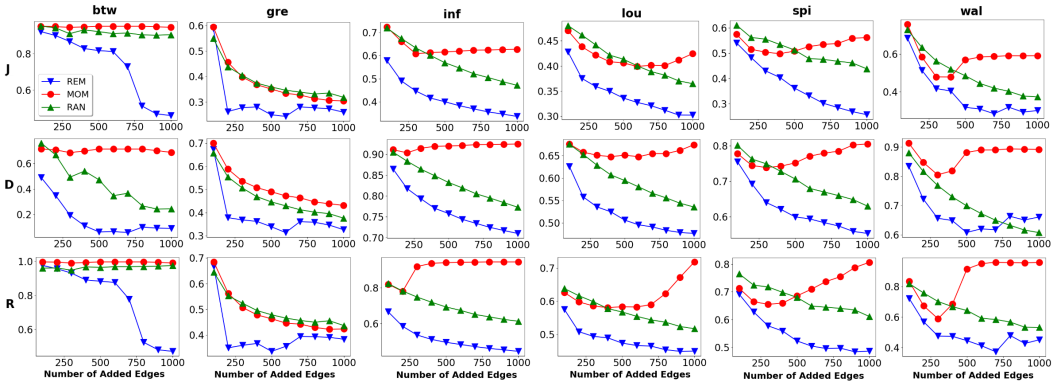

Figure 2: Compare the deception effect of REM, MOM and RAN for dataset Eml.

**Result set 1.** Fig. 1 plots the the values of scores $J, D, R$ (for communities detected by Louvain) as the normalized residual entropy increases. The three scores unanimously increase with $\rho_{\mathcal{P}}$, validating our intuition that $\rho_{\mathcal{P}}$ can be used to obfuscate $\mathcal{P}$. Moreover, the correlations are almost linear for Fbk, Pow and PGP.

**Result set 2.** We then examine the performance of the three deceptors over 9 data sets when adding a budget $k$ edges. Due to varying graph sizes, we set $k = 20$ for Dol; $k = 1000$ for Jaz, Eml, Fbk,

Pow; $k = 2000$ for PGP; $k = 10000$ for CAI, Bri; and $k = 20000$ for Liv. Table 2 compares the $J, D, R$ scores for different algorithms. Clearly, REM performs better than MOM and RAN in almost all scenarios. For Louvain & SpinGlass, REM gives the unanimous best results across all datasets and scores. The recall $R$ for most cases are less than or close to 0.5 for REM which is not true for the other two deceptors. On the other hand, with a small budget percentage $k/|E|$ for larger graphs, REM can achieve better community deception, which means that the advantage of REM becomes more prominent for larger graphs. Fig. 2 shows the trend of $J, D, R$ scores as the number of added edges increases for 6 community detectors over the data set Eml. Among the three deceptors $\{\text{MOM}, \text{REM}, \text{RAN}\}$, MOM has the worst performance. The only case that MOM performs better is for the detector gre, which is based on a greedy modularity maximization strategy. Overall, REM achieves the best anonymity in all the six detectors. In particular, under the three algorithms $\{\text{inf,lou,spi}\}$, the $J, D, R$ scores consistently decrease. These results validate REM's effectiveness in hiding community structures.

**Result set 3.** Finally, we check the preservation of the data after applying REM. First, by implementing the REM algorithm, we reduce the Jaccard index to below $0.5$ for CSD. We then check the changes of some key indicators, i.e., clustering coefficient (CC), mean shortest path length (MSPL), and the percentage of nodes with the top-10% Pagerank and Betweenness after applying REM. As shown in Table 3, these indicators has no significant change due to applying REM. In general, a larger network leads to less change. Among them, for Fbk, 10%-PageRank and 10%-betweenness change greatly. This is because the vertices in this data set tend to have very similar PageRank and betweenness scores; Since Pow represents a large-scale power grid, it naturally has a large mean shortest path length. This value will shift greatly when more links are created between the communities.

Table 3: The changes of some key indicators after applying REM.

| Data | $|E'|$ | Jaccard | CC | MSPL | $10\%-$ Pagerank | $10\%-$ Betweenness |
|------|------|---------|-----|------|------------------|---------------------|
| Dol | 10 | $1 \rightarrow 0.44$ | $0.308 \rightarrow 0.298$ | $3.357 \rightarrow 2.996$ | $1 \rightarrow 0.833$ | $1 \rightarrow 0.833$ |
| Jaz | 250 | $1 \rightarrow 0.48$ | $0.520 \rightarrow 0.498$ | $2.230 \rightarrow 2.070$ | $1 \rightarrow 0.895$ | $1 \rightarrow 0.842$ |
| Eml | 100 | $1 \rightarrow 0.39$ | $0.166 \rightarrow 0.166$ | $3.606 \rightarrow 3.577$ | $1 \rightarrow 0.991$ | $1 \rightarrow 0.982$ |
| Fbk | 550 | $1 \rightarrow 0.48$ | $0.0004 \rightarrow 0.0004$ | $3.867 \rightarrow 3.539$ | $1 \rightarrow 0.243$ | $1 \rightarrow 0.118$ |
| Pow | 200 | $1 \rightarrow 0.49$ | $0.103 \rightarrow 0.101$ | $18.99 \rightarrow 13.70$ | $1 \rightarrow 0.953$ | $1 \rightarrow 0.644$ |
| PGP | 400 | $1 \rightarrow 0.45$ | $0.378 \rightarrow 0.377$ | $7.485 \rightarrow 7.279$ | $1 \rightarrow 0.979$ | $1 \rightarrow 0.930$ |
| CAI | 1000 | $1 \rightarrow 0.49$ | $0.007 \rightarrow 0.007$ | $3.875 \rightarrow 3.869$ | $1 \rightarrow 0.983$ | $1 \rightarrow 0.977$ |
| Bri | 1000 | $1 \rightarrow 0.44$ | $0.111 \rightarrow 0.111$ | $4.858 \rightarrow 4.854$ | $1 \rightarrow 0.993$ | $1 \rightarrow 0.971$ |

## 5 Conclusions and Future Work

In this paper, we introduce the community structure deception (CSD) problem, utilize community based structural entropy to the CSD problem, and propose a residual minimization (REM) algorithm. We reduce search space to critical edges to optimize REM, which allows our community structure deceptor to run very efficiently. Experiments show that our algorithm REM performs better than RAN and MOM in almost all attack scenarios.

Some potential directions of future work include (1) extending the method to hide communities in weighted and directed graphs; (2) investigating the problem of hiding overlapping community structures; (3) hiding other structural properties, e.g., influential nodes, hierarchies, etc. and (4) explore the connection between structural entropy and community detection.

**Acknowledgments**

This work is supported by Provincial Key Research and Development Program of Zhejiang (Grant No. 2019C03133) and Major Scientific Research Project of Zhejiang Lab (Grant No. 2018FD0ZX01). The co-authors Angsheng Li and Jiamou Liu are supported by the National Natural Science Foundation of China (No. 61932002). We also thank our anonymous reviewers for their constructive comments.

## Footnotes

*is a professor in the School of Computer Science and Technology, Beijing Institute of Technology. He is selected into the Program for New Century Excellent Talents in University from Ministry of Education, China. His research interests include Internet of things, cloud computing security, and blockchain.

[2]If $E'$ is $\{e\}$, we abuse the notation writing $G \oplus e$ for $G \oplus \{e\}$.

[3]Trials are conducted on a Server *Xeon(skylake) platnum 8163 cpu 2.5GHz (12 cores, non-parallel computing)* and *16GBs RAM*

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
