[Supplementary Material]

# REM: From Structural Entropy To Community Structure Deception

Yiwei Liu[1,2], Jiamou Liu[3], Zijian Zhang[1,3], Liehuang Zhu[1], Angsheng Li[4]

[1]Beijing Institute of Technology, Beijing, China;  [2]Zhejiang University, Zhejiang, China

[3]The University of Auckland, Auckland, New Zealand;  [4]Beihang University, Beijing, China

{yiweiliu, zhangzijian, liehuangz}@bit.edu.cn, jiamou.liu@auckland.ac.nz, angsheng@buaa.edu.cn

## Background

The disclosure of the users' community affiliations leads to privacy leak. This raises the problem of **community structure deception (CSD)**, which asks for ways to minimally modify the network so that a given community structure maximally hides itself from community detection algorithms.

## Problem Formulation

☐ A social network $\longrightarrow$ An undirected connected graph $G = (V, E)$.

☐ The community structure $\longrightarrow$ a partition $P = \{X_1, X_2, \cdots, X_L\}$ of $V$.

☐ obfuscating $P \longrightarrow$ by **adding a number of "dummy edges"**.

We measure the similarity between $P$ and $P'$ by three metrics:

➤ $J(P, P')$: $J$ is *jaccard index*

➤ $D(P, P')$: $D$ is *normalized mutual information*

➤ $R(P, P')$: $R$ is the *recall*

## Residual Entropy-based CSD

**Definition 1. (Shannon entropy)** $\mathcal{H}(G)$ captures the average number of bits needed to encode the $n$ vertices in a lossless way:

$$\mathcal{H}(G) := -\sum_{i=1}^{|V|} \frac{d_i}{2|E|} log_2 \frac{d_i}{2|E|},$$

where $d_i$ is the degree of vertex $i$.

A: 000  B: 001  C: 10    D: 11  E: 010  F: 011    $\mathcal{H}(G) = 5.11$

**Definition 2. The structural entropy of $G$ relative to $P$ is:**

$$\mathcal{H}_P(G) := -\sum_{j=1}^{L} \frac{v_j}{2|E|} \mathcal{H}(G \restriction X_j) - \frac{g_j}{2|E|} log_2 \frac{v_j}{2|E|},$$

where $\mathcal{H}(G \restriction X_j) := \sum_{i \in X_j} \frac{d_i}{v_j} log_2 \frac{d_i}{v_j}$ and $g_j$ denotes the number of edges with exactly one end point in $X_j$.

A: 00  B: 01  C: 1    D: 1  E: 00  F: 01    $\mathcal{H}_P(G) = 3.40$

**Definition 3. The normalized residual entropy of $P$ is**

$$\rho_P(G) := (\mathcal{H}(G) - \mathcal{H}_P(G))/\mathcal{H}(G).$$

## REM: Algorithm and optimization

**Theorem 1.** There exists a critical non-edge $\{u, v\}$ is RE-minimizing.

**Theorem 2.** Examining the critical edges implements in $\mathcal{O}(L|V|)$.

---

**Algorithm 1:** An efficient REM deceptor

**Input:** Graph $G = (V, E)$, $\mathcal{P} = \{X_1, X_2, \ldots, X_L\}$
**Output:** A non-edge $\{u^*, v^*\}$

1  Initialize $\rho^* \leftarrow 1$;
2  **for** $s \leftarrow 1$ to $L$ and $t \leftarrow s$ to $L$ **do**
3      **if** $X_s \times X_t$ contains no non-edge **then**
4          continue;
5      **for** all critical non-edge $\{u, v\}$ in $X_s \times X_t$ **do**
6          Set $\rho_{u,v} \leftarrow (\mathcal{H}(G \oplus \{u, v\}) - \mathcal{H}_\mathcal{P}(G \oplus \{u, v\}))/\mathcal{H}(G \oplus \{u, v\})$;
7          **if** $\rho_{u,v} < \rho^*$ **then**
8              Set $\rho^* \leftarrow \rho_{u,v}$, $u^* \leftarrow u$, and $v^* \leftarrow v$;

9  **return** $\{u^*, v^*\}$;

---

A crude implementation runs in $\mathcal{O}(|V|^2)$ time. We present an $\mathcal{O}(L|V|)$-implementation by only examining the critical edges.

## Experiments

**Dataset:** 9 real-world networks. **Adversary:** $\{btw, gre, inf, lou, spi, wal\}$.
**Evaluation:** 3 metrics, $\{J, D, R\}$. **Benchmark:** $\{MOM, RAN\}$.

The preservation of the data after applying REM:

| Dataset | $|E'|$ | Jaccard | Clustering coefficient | Mean shortest path length | 10% Pagerank | 10% Betweenness |
|---------|------|--------------------|-----------------------|---------------------------|----------------------|----------------------|
| Dol | 10 | $1 \rightarrow 0.44$ | $0.308 \rightarrow 0.298$ | $3.357 \rightarrow 2.996$ | $1 \rightarrow 0.833$ | $1 \rightarrow 0.833$ |
| Jaz | 250 | $1 \rightarrow 0.48$ | $0.520 \rightarrow 0.498$ | $2.23 \rightarrow 2.070$ | $1 \rightarrow 0.895$ | $1 \rightarrow 0.842$ |
| Eml | 100 | $1 \rightarrow 0.39$ | $0.166 \rightarrow 0.166$ | $3.606 \rightarrow 3.577$ | $1 \rightarrow 0.991$ | $1 \rightarrow 0.982$ |
| PGP | 400 | $1 \rightarrow 0.45$ | $0.378 \rightarrow 0.377$ | $7.485 \rightarrow 7.279$ | $1 \rightarrow 0.979$ | $1 \rightarrow 0.930$ |
| CAI | 1000 | $1 \rightarrow 0.49$ | $0.007 \rightarrow 0.007$ | $3.875 \rightarrow 3.869$ | $1 \rightarrow 0.983$ | $1 \rightarrow 0.977$ |
| Bri | 1000 | $1 \rightarrow 0.44$ | $0.111 \rightarrow 0.111$ | $4.858 \rightarrow 4.854$ | $1 \rightarrow 0.993$ | $1 \rightarrow 0.971$ |

## Conclusions

➤ Utilize community based structural entropy to the CSD problem

➤ Propose a residual minimization (REM) algorithm.

➤ Reduce search space to critical edges to optimize REM.

➤ Validate the performance of our algorithm.