[Reviews · NeurIPS 2019]

Reviewer 1



The paper introduces a new interesting problem: to add a fixed number of edges to obfuscate the community structure of a given graph as much as possible. This problem is different from existing problems that focus on the deception of a single community, and it is related to mitigate privacy risks caused by community detection. While the proposed algorithm simply chooses edges greedily, it is novel in two aspects. First, it is based on a new and understandable measure for the strength of community structure. The paper shows experimentally that minimizing the new measure better obfuscates community structure than minimizing modularity, which is a widely-used measure. Second, the algorithm is carefully designed so that it takes only O(#communities \times #nodes) time. The experimental evaluation of the proposed algorithm is extensive. 9 datasets and 6 community-detection methods are used to show that the proposed algorithm outperforms baseline methods consistently regardless of datasets and community detection methods. However, the two baselines methods used are very weak. One (RAN) is simple random edge addition, and the other one (MOM), which greedily minimizes modularity, surprisingly underperforms RAN. Why MOM underperforms RAN is unclear. Greedy algorithms for minimizing variants of modularity or other measures can be considered as baselines. The paper is well written and easy to follow.

Reviewer 2



This paper considers the problem that information about the community structure of a network, together with information on some nodes, can be used to infer information about other nodes. Thus creating a potential privacy concern. To remedy this one would like to be able to hide the true community structure in a way that does not change the network too much. In this paper, the authors give a formal formulation for this problem. They then define a class of deception algorithms, that would hide the community, and ways to measure how well such an algorithm performs. They then develop a community deception algorithm, based on entropy measures of edges. Their main result is that this algorithm runs is time that, given the number of communities, is linear in the number of nodes. Experiments are conducted to show the algorithm runs fast. In addition, the authors test how well it can deceive six community detection algorithms. Their algorithm is shown to outperform both the base line algorithm, which randomly adds edges between communities, and one which adds edges that minimize modularity. I think the problem this paper tries to address is quite relevant and also interesting from a theoretical perspective. The solution proposed in this paper, adding edges that reduce the information contained in the community structure, seems to me like a very nice approach. The experiments do definitely show that this deception algorithm performs well. Unfortunately, I had some problems with the main theoretical results regarding the running time. In particular, both the standard and the efficient algorithm have a statement that checks, for every pair of communities, if there are nodes in different communities that do not have an edge (a non-edge). Unless some additional information is encoded in the data structure, finding all non-edges has running time |V|^2. The authors do mention that looping over all critical non-edges can be done efficiently, using specifically designed data structures. Unfortunately, I did not understand why this would imply that it can be done in linear time, since there might be a quadratic number of edges. Either I am missing something obvious, there are some details that are not explained, or the result is not true. I believe it is the second option, but I can’t extract the needed information from the text. I would have been nice if I could have looked at the code, but there was no code available in the submission package, nor a link where the code could be downloaded. Even though it is stated in the Reproducibility Response that it should be. I also found several pieces of text that I was unable to understand. This was especially true for Section 3, where the authors explain their algorithm. What also didn’t help was that some objects and expressions where introduced and used, without a proper definition being given. These things made it difficult for me to really understand the innovation in the deception algorithm and the intuition behind it. To summarize, the paper tries to tackle a relevant problem using an interesting idea, whose performance is positively verified by experiments. However, there are several parts of the paper that are very unclear, most notably the text on the new algorithm and the proof of the main theorem. In addition, the Supplementary Material felt lacking in conclusions and proper explanation. Based on this I can only place this submission marginally above the acceptance threshold. ----------------------------------------------------- After rebuttal: The authors addressed my concerns regarding the computational complexity but I still believe the result as stated is somewhat misleading since it assumes a very specific data structure. I really think this point should be made crystal clear when revising the manuscript. Given that the other recommended adjustments will be included in the revised version I will keep my recommendation. ----------------------------------------------------- Below is a more detailed list of comments: Main text Line 46 […schemes of the…] I think this should be “schemes with the” Line 57 […connected between them…] This should be “connected among each other” Line 66 [...random walk…] This should be plural “random walks” Line 68 […numbers bits…] This should be “number of bits”. Also, I believe an article should be included at the end “the random walk”. Line 94 [Suppose \mathcal{P}^\prime be…] This should be “Suppose \mathcal{P}^\prime is” Line 102 [Define…as [25]]. This was the reader has to first go and look at [25] before he/she can continue reading your paper. I would strongly recommend to include the definitions here, to make the main text self-contained. Line 103 [information is then…] I would suggest writing “information is then defined as” Line 104 […utilizes the range [0,1] well] What does this mean? I never encountered a statement about a measure that utilizes a given range well. Please explain. Line 129 [In actual fact, however,…] I would simply write “However,” Lines 142-149 [The behavior…receiver as follow:] I was unable to understand what is explained here. One reason for this could be the structure of the sentences, which made it hard to grasp their meaning. Another reason is the sudden injection of terminology. For example, I did not understand what a codeword is in the last sentence and how it relates to the rest of the text. Please have a good look at the text, including sentence construction and spelling, and rewrite it so that it is more clear to the reader. Line 151 […lossless way.] Please explain what this means. Lines 153-167 [\mathcal{H}(G) merely…entropy measure] Similar to my comment above, this piece of text was very difficult to understand. An important role here is played by the “codeword” which was introduced in the paragraph above but never properly explained. Again, grammar and sentence construction are also part of the confusions. An example of this is the sentence “This wil not…network G.”. This is crucial part of the paper since it explains the main intuition behind your approach. So please rewrite this part so that it is clear. Equation (3): The term \mathcal{H}(G|X_j) is never defined. I think it is the conditional entropy of G given X_j, but it would help the reader if this is mentioned in the text. Lines 175-176 […that \mathcal{P} brings…] This should be “that \mathcal{P} contains”. Also, [..harder to be detected] should be “harder to detect” Line 184 [A non-edge is…] I would suggest writing “non-edge is a pair”. Lines 185-187 […the least degree…] I think that technically this is correct. However, since we are talking about a measured quantity here, it feels that “smallest” is more appropriate. Therefore, I would suggest to write “the smallest degree” of “the minimum degree”. There are two other instances of the word least being used in the next two lines and one more on line 206. Line 192 […F is monotonic…] Shouldn’t you state that is monotonic increasing? Line 199 [For any communities…] I would write “For any two communities” Line 201 What do you mean with Y \sim (v_1/2|E|, …, v_L/2|E|)? I think it means that Y is selected uniformly at random from this set of values. However, the way it is written here is not standard notation in probability theory. I would suggest writing “Let Y be sampled uniformly at random from (v_1/2|E|, …, v_L/2|E|)”. The same of course goes for Z. Lines 217-226: As mentioned in my review above, this proof was very unclear to me. One obvious question I had is why it is L and not L^2? We are going over all pairs among L communities. Moreover, it was not clear at all to me why the second loop lines 5-8 can be done L|V| time. Please expand this proof to include all these details. Also, it would be helpful if you supplied the code where this algorithm (or its efficient version) is implemented. Lines 228-229: What does X_s & X_t mean? Lines 239-240 [To validate…resembles Alg 1….] Can you explain how this edge is selected? Is it at random or according to some selection criteria? Also, what does “crude” mean here? Can you really talk about a crude implementation when you are only computing it for a single given edge? Line 245 […in the decreasing…] This should be “in decreasing” Table 2: Overall, when the REM algorithm is not the best, it still seems to perform on par with the best one. However, this is not the case for the Dol network, where RAN is the best (0.18) and REM is much larger (0.38). Can you explain why this happened? Line 270 […is not held true…] This should be “is not true” Supplementary Material Lines 6-10: This text is difficult to understand. Please rewrite the text, carefully considering the sentence construction and grammar. Section II: It is completely unclear to me why this section is included, and this is never explained. Not in the main text and not here. Are you using a different algorithm for MOM then what is done in the literature? If so, then it would help to reader if you mention this. This would also add more weight to the proofs. Please add more explanation to this section, explaining why it is included and why you need to derive results for the algorithm. Line 66 […chooses a non-edges…] This should be “chooses non-edges” Line 70 […begin and… adding some certain mount…] This should be “beginning and…adding a certain amount” Line 71 […with the increasing of the number of adding edges…] I think this should be “with increasing number of added edges” Line 72 […MOM with adding…] This should be “MOM after adding” Line 92 [We have known that…] I think this should either be “We know that” or “We have shown that” (if you refer to the part in the main text where this is done) Line 103 [line 2, We…] This should be “line 2, we” Lines 107-108 [For example…datasets] I feel that some conclusion is missing here. What do figures 4 and 5 tells us about the deception algorithms? Line 110 […over community with…] This should be “on communities with” Line 124 […A generating small-world…] This should be “Generating a small-world” Line 132 [Fig 6 shows…scores] Why do you not include R in these figures? Line 134 [To LFR…] This should be “For LFR” Line 135 […to small world…] This should be “for the small-world” Line 136 […the small world structure…] What do you mean with small-world structure? Does this involve the scaling of the average shortest path length or the diameter or something else? This is not well-established terminology. Fig 6. Why aren’t these results compared with the baseline RAN?

Reviewer 3



This paper proposed a method REM for community structure deception, aiming to avoid privacy leaking in community detection. This work is motivated from an important application issue. However, advantages of their proposed method are not clear. Rather than proofs on running time, this paper should discuss more on (1) why and how much REM improves on hiding sensitive network structure, (2) how this avoids privacy leaking (an example in main paper rather than supplementary would help), and (3) how this would affect us to extract useful, non-sensitive information from network. One question should be discussed: Can this method hide block membership for all kinds of nodes including those high degree nodes in cluster centers? Or only peripheral nodes on cluster boundaries ( which could be a great amount)?

[Author Response · NeurIPS 2019]

We thank the reviewers for the work. **To Reviewer 1: (a)** $\mathcal{H}(G\restriction_{X_j})$ is the result of the $\mathcal{H}$ function on the induced subgraph $G\restriction_{X_j}$ of $G$. **(b)** As suggested, we tested our results on a variants of modularity QDS as baseline [1]. Fig.(left) confirms that REM still performs better over datasets Jaz and Eml. **(c)** To explain the inferior performance of MOM compared to RAN, we remark that modularity is an index easily affected by community size [2]. MOM will most likely add edges to two communities with the largest volume (See Lem. 2 in Supplementary II). We also gave explanation at the end of Sec. 2 in the main paper. **To Reviewer 2: (a)** Due to restriction to use url, we cannot share code that shows detailed implementation. The $O(|V|L)$ complexity can be justified as follows: For communities $X_i$, $X_j$, suppose a data structure is used that assigns to each node $x \in X_i$ the node $x' \in X_j$ where no edge exists between $x$ and $x'$, and $x'$ is such a node with minimum degree. To find the desired critical edge, the algorithm may scan over all such pairs $(x, x')$ where $x \in X_i$. This takes $O(|X_i|)$. Similarly, the algorithm examines over all pairs $(y, y')$ where $y \in X_j$ and $y' \in X_i$ is defined analogously as $x'$. This takes $O(|X_j|)$. Hence, for $X_i, X_j$ the algorithm takes $O(|X_i| + |X_j|)$. Thus, for any $X_i$, the algorithm will take $O(L|X_i| + |X_1| + \cdots + |X_L|) = O(L|X_i| + |V|)$. The overall time takes $O(L|X_1| + \cdots + L|X_L| + L|V|) = O(L|V|)$. The implementation of the required data structure would store for each node $x \in X_i$, collections of nodes $Y_d, Y_{d+1}, \ldots, Y_g \subseteq X_j$ where $Y_d$ contains all nodes $y \in X_j$ such that $\{x, y\}$ is a non-edge and $y$ has degree $d$, where $d, g$ are the least and greatest integers where $Y_d, Y_g$ are non-empty. This makes sure that the data structure can be built and updated in the required time complexity. **(b)** Section 3 motivates structural entropy from a data communication perspective, to stay consistent with the origin of information theory. Our entropy notion measures the cost of communication between nodes in the network that is inherent to network topology. We view each node as a communication station that is able to pass message to adjacent nodes. To send a message, the sender needs to give an "address" of the receiver node, which is a string in a fixed (say binary) alphabet. This is the so-called *codeword*, which is used to locate a node in the network. The structural entropy $\mathcal{H}(G)$ is the expected length of the codeword when we assume that the messages are passing between randomly chosen node to a randomly chosen neighbor. **(c)** "Lossless way" refers to a data compression algorithm that allows the original data to be perfectly reconstructed from the compressed data. **(d)** The size of dataset Dol is very small with only 62 vertices. Adding 100 random edges will drastically change the original community structure which explains the good performance of RAN. **To Reviewer 3. (a)** REM aims at minimizing $\rho_{\mathcal{P}}(G)$. See Fig.(right) as an example of edge adding which distorts the community structure of a graph. To measure how much privacy is leaked, we use the normalized mutual information $D$ between structure (a) and (b). Thus we may view $1 - D$ as an indication of how much sensitive network structure is protected. **(b)** Affiliation relation disclosure can lead to serious privacy leaking. E.g., Wondracek et al. in [3] showed that information about the community memberships of a user (i.e., the groups of a social network to which a user belongs) is sufficient to uniquely identify this person, or, at least, to significantly reduce the set of possible candidates. In [4], communities are used to re-identify multiple addresses belonging to a same user in Bitcoin trading networks. **(c)** By adding a fixed (small) number of edges, we aim to minimize the change to other "insensitive" information of a network. The table lists changes to the clustering coefficient, mean shortest path length, the percentage of nodes with the top-10% Pagerank and Betweenness after applying our algorithm.

| Dataset | $|E'|$ | Jaccard | Clustering coefficient | Mean shortest path length | 10% Pagerank | 10% Betweenness |
|---|---|---|---|---|---|---|
| Dol | 10 | $1 \to 0.44$ | $0.308 \to 0.298$ | $3.357 \to 2.996$ | $1 \to 0.833$ | $1 \to 0.833$ |
| Jaz | 250 | $1 \to 0.48$ | $0.520 \to 0.498$ | $2.23 \to 2.070$ | $1 \to 0.895$ | $1 \to 0.842$ |
| Eml | 100 | $1 \to 0.39$ | $0.166 \to 0.166$ | $3.606 \to 3.577$ | $1 \to 0.991$ | $1 \to 0.982$ |
| PGP | 400 | $1 \to 0.45$ | $0.378 \to 0.377$ | $7.485 \to 7.279$ | $1 \to 0.979$ | $1 \to 0.930$ |
| CAI | 1000 | $1 \to 0.49$ | $0.007 \to 0.007$ | $3.875 \to 3.869$ | $1 \to 0.983$ | $1 \to 0.977$ |
| Bri | 1000 | $1 \to 0.44$ | $0.111 \to 0.111$ | $4.858 \to 4.854$ | $1 \to 0.993$ | $1 \to 0.971$ |

[1] Chen, Kuzmin, Szymanski. Community Detection via Maximization of Modularity and Its Variants. 2014.

[2] Fortunato Santo. Community detection in graphs. 2009.

[3] Wondracek G , Holz T , Kirda E , et al. A Practical Attack to De-anonymize Social Network Users. 2010.

[4] Remy, Rym, Matthieu. Tracking bitcoin users activity using community detection on a network of weak signals. 2017.



[Meta-Review · NeurIPS 2019]

This submission deals with privacy issues of revealing community information in graphs and ways to hide this information by modifying the network structure. There is consensus that the methods proposed in the submission are effective for the task of hiding community membership and that the problem is interesting from both a theoretical and practical perspective. For these reasons, I think the paper should be accepted. The reviewers raised some ways in which the paper could be improved. One specific issue is what types of nodes get their attributes hidden. For example, are these "core" nodes in the community with high degree or low-degree periphery nodes? A deeper investigation into how the method gets good performance would be much appreciated.